# Full-gap superconductivity in spin-polarised surface states of topological semimetal $\beta$-PdBi$_2$

K. Iwaya [1], Y. Kohsaka[1], K. Okawa[2], T. Machida[1], M.S. Bahramy[1,3], T. Hanaguri[1] & T. Sasagawa[2]

A bulk superconductor possessing a topological surface state at the Fermi level is a promising system to realise long-sought topological superconductivity. Although several candidate materials have been proposed, experimental demonstrations concurrently exploring spin textures and superconductivity at the surface have remained elusive. Here we perform spectroscopic-imaging scanning tunnelling microscopy on the centrosymmetric super-conductor $\beta$-PdBi$_2$ that hosts a topological surface state. By combining first-principles elec-tronic-structure calculations and quasiparticle interference experiments, we determine the spin textures at the surface, and show not only the topological surface state but also all other surface bands exhibit spin polarisations parallel to the surface. We find that the super-conducting gap fully opens in all the spin-polarised surface states. This behaviour is con-sistent with a possible spin-triplet order parameter expected for such in-plane spin textures, but the observed superconducting gap amplitude is comparable to that of the bulk, sug-gesting that the spin-singlet component is predominant in $\beta$-PdBi$_2$.

[1] RIKEN Center for Emergent Matter Science, Wako, Saitama 351-0198, Japan. [2] Laboratory for Materials and Structures, Tokyo Institute of Technology, Yokohama, Kanagawa 226-8503, Japan. [3] Department of Applied Physics, The University of Tokyo, Hongo, Bunkyo-ku, Tokyo 113-8656, Japan. Correspondence and requests for materials should be addressed to K.I. (email: iwaya@riken.jp) or to T.H. (email: hanaguri@riken.jp)

Superconductivity arising from non-degenerate spin-polarised Fermi surfaces is expected to consist of a mixing of spin-singlet and triplet order parameters as proposed in non-centrosymmetric superconductors[1, 2]. If superconductivity is induced in the spin-polarised topological surface states (TSSs), such a situation may be realised and the induced spin-triplet order parameter could play a key role for the emergence of Majorana fermions at edge states or in vortex cores, offering potential applications such as topological quantum computing[3–6]. To accomplish topological superconductivity, various candidate systems including carrier doped topological insulators[7–9] and superconductor/topological insulator heterostructures[10–14] have been investigated, in addition to other systems such as magnetic[15] and semiconducting[16] nanowires fabricated on superconductors.

Among these systems, stoichiometric superconductors possessing the TSSs at the Fermi level $E_F$ are promising candidates. The non-centrosymmetric superconductor PbTaSe₂[17] is one of such materials for which a recent spectroscopic-imaging scanning tunnelling microscopy (SI-STM) study has revealed the existence of both the TSS and a fully opened superconducting (SC) gap[18]. At the cleaved surface of PbTaSe₂, the inversion symmetry is broken not only along the surface normal direction but also along the in-plane direction because of the non-centrosymmetric crystal structure. This causes intricate spin-orbit coupling that gives rise to out-of-plane spin components. The SC order parameter favoured in such a situation is interesting but complicated.

Here we study another stoichiometric superconductor $\beta$-PdBi₂ with a SC transition temperature $T_c \sim 5.4$ K[19]. $\beta$-PdBi₂ crystallises into centrosymmetric tetragonal lattice with space group I4/mmm. Since in-plane inversion symmetry is always maintained, the spin texture induced by spin-orbit coupling is expected to be aligned in the in-plane direction at the surface. Angle-resolved photoemission spectroscopy (ARPES) has revealed the presence of the TSS as well as the topologically trivial surface state at $E_F$[20]. The SC gap has been studied by scanning tunnelling spectroscopy experiments[21, 22]. However, reported tunnelling spectra do not agree with each other; there is only one BCS-like gap at the surface of the bulk single crystal cleaved in the air[21], whereas two gaps are identified in the case of the in situ molecular-beam-epitaxy-grown thin film[22]. Moreover, the simultaneous investigation of the TSS and the SC gap, which is critically important to discuss topological superconductivity, has not yet been done. We have performed SI-STM on clean surfaces of $\beta$-PdBi₂ prepared by cleaving bulk single crystals in ultra high vacuum to study the SC gap opening in the TSS. Quasiparticle interference (QPI) imaging combined with numerical simulations has been utilised to explore the spin textures at the surface. We find that the observed QPI patterns are qualitatively consistent with the spin textures predicted by the first-principles calculations, which reveal that not only the TSS but also all other states at the surface are spin polarised. This suggests that the surface of $\beta$-PdBi₂ can potentially harbour the mixed spin-singlet and triplet superconductivity. A fully opened single SC gap $\Delta \sim 0.8$ meV has

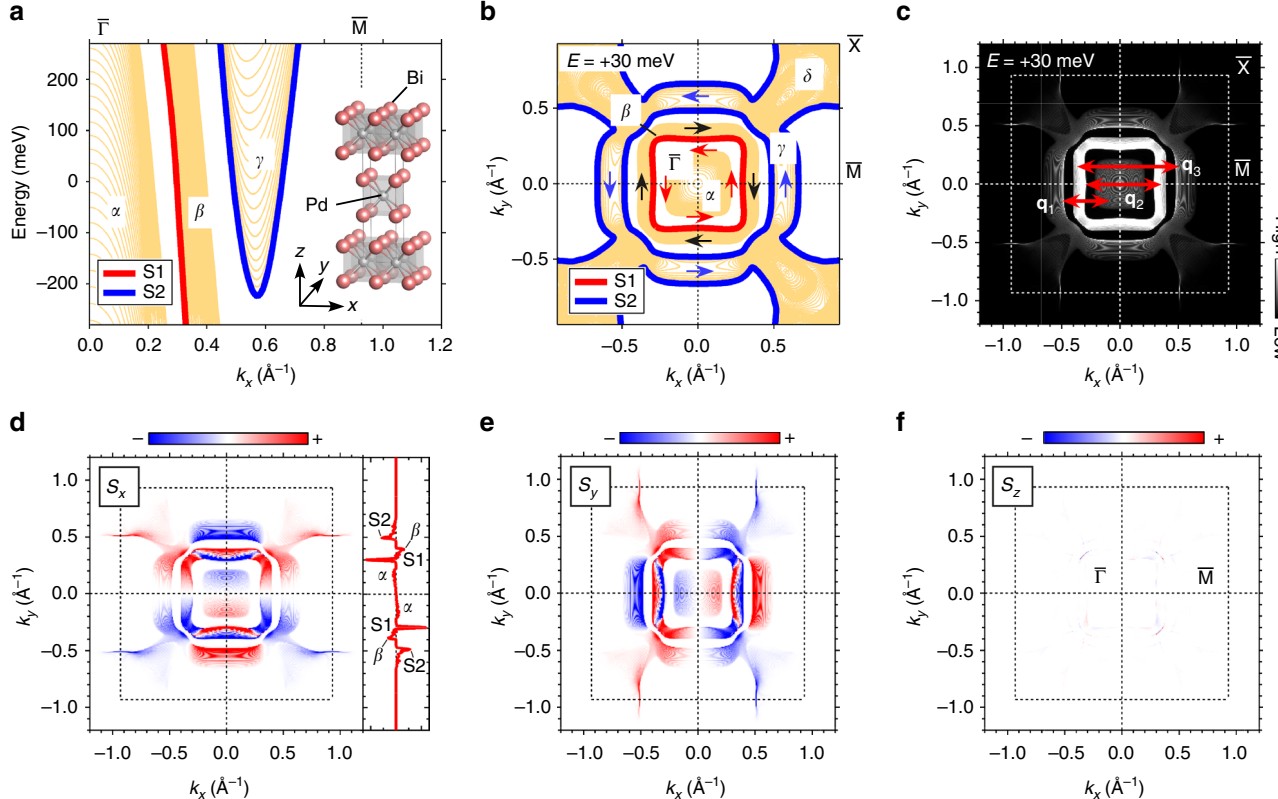

**Fig. 1** Calculated surface electronic states of the top-most Bi layer of $\beta$-PdBi₂. **a** Energy dispersions of the surface (topologically trivial S1 and nontrivial S2) and the bulk states ($\alpha$, $\beta$ and $\gamma$) along the $\overline{\Gamma}-\overline{M}$ direction. The inset shows the crystal structure of $\beta$-PdBi₂. **b** The constant energy contours (CECs) at energy $E = +30$ meV. The CECs consist of the two surface states (S1 and S2) and the bulk states ($\alpha$, $\beta$, $\gamma$ and $\delta$). The spin orientations of S1 (red), S2 (blue) and $\beta$ (black) are schematically shown by arrows. **c** Spectral function at $E = +30$ meV with the Wannier transformation written in Methods section. The interband forward-scattering from $\alpha$ to $\overline{\Gamma}$-centred S2, the inter-band back-scattering from S1 to $\beta$, and from $\overline{\Gamma}$-centred S2 to $\beta$ along the $\overline{\Gamma}-\overline{M}$ direction is indicated as $q_1$, $q_2$ and $q_3$, respectively. A dashed square denotes the first Brillouin zone. **d–f** Spin polarisations weighted by the spectral function at $E = +30$ meV are shown for all components, $s_x$, $s_y$ and $s_z$. A line profile of $s_x$ along $\overline{M}-\overline{\Gamma}-\overline{M}$ is shown in **d** where S1, S2, $\alpha$ and $\beta$ are clearly identified. The spin orientations of S1, S2 and $\alpha$ are anticlockwise whereas the spin orientation of the $\beta$ band is clockwise

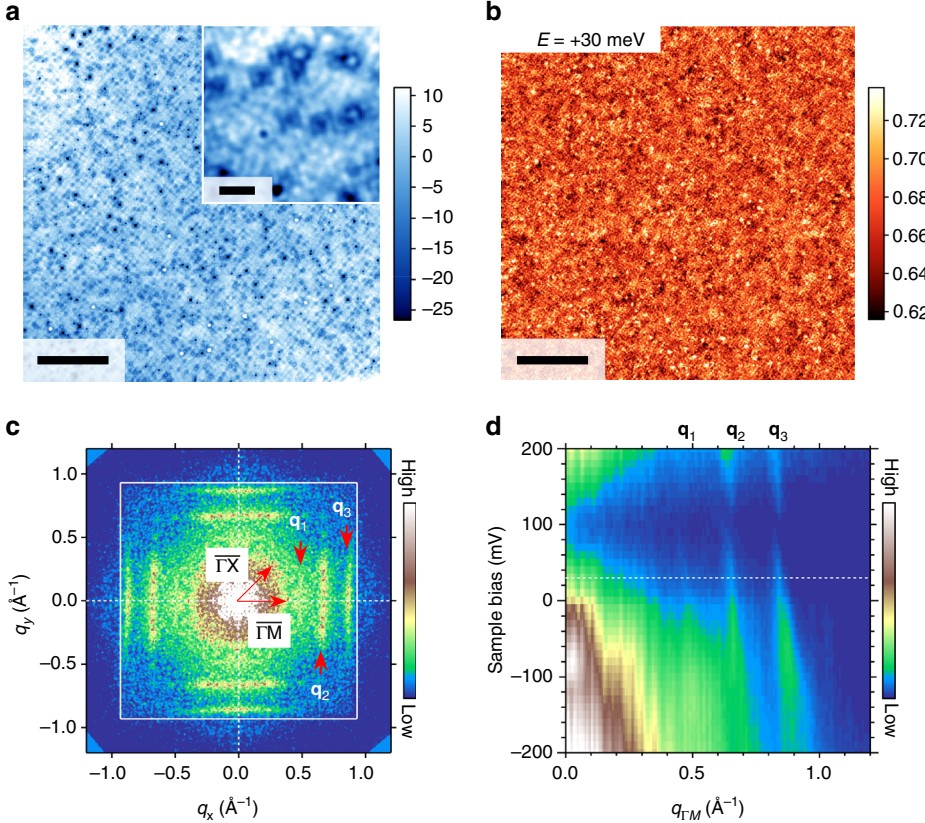

**Fig. 2** Spectroscopic imaging on the $\beta$-PdBi$_2$ surface. **a** A constant-current STM image of the $\beta$-PdBi$_2$ surface prepared by cleaving in ultra high vacuum. Set-up conditions for imaging were $V = +200$ mV, $I = 200$ pA. The image was taken at temperature $T = 1.5$ K. The scale bar corresponds to 20 nm and the colour scale is in pm. A magnified image of the small area ($10 \times 10$ nm$^2$) is shown in the inset. The scale bar corresponds to 2 nm. **b** d$I$/d$V$ map at $V = +30$ mV taken at the same field of view as **a**. Set-up conditions for imaging were $V = +200$ mV, $I = 200$ pA. For spectroscopic measurements, bias voltage was modulated at 617.3 Hz with an amplitude $V_{r.m.s.}$ of 3.5 mV. The modulation amplitude is larger than the SC gap and thus the effect of superconductivity is not clear at 0 mV. Data were taken at $T = 1.5$ K. The scale bar corresponds to 20 nm and the colour scale is in nano siemens (nS). **c** Fourier-transformed image of **b**. Three parallel line-like QPI signals perpendicular to the $\overline{\Gamma}-\overline{M}$ direction are labelled as $q_1$, $q_2$ and $q_3$. The image is symmetrised with respect to the fourfold symmetry of the crystal structure and rotated for clarity. A white square denotes the first Brillouin zone. **d** Line profile from the series of Fourier-transformed d$I$/d$V$ maps taken at different $E$ along the $\overline{\Gamma}-\overline{M}$ direction. Dispersions of $q_1$, $q_2$ and $q_3$ are clearly seen. QPI signals around +100 mV are suppressed due to the set-point effect (Supplementary Note 1)

been identified in the tunnelling spectrum without any detectable residual spectral weight near $E_F$, being consistent with the result of air-cleaved sample[21]. These results indicate that the electronic states detected by SI-STM including the TSS are all spin-polarised and exhibit fully gapped superconductivity. We argue that the spin-triplet order parameter expected from the spin textures is consistent with the observed isotropic gap. The observed gap amplitude, however, is comparable to that of the bulk[23], suggesting that the spin-singlet component is dominant. Therefore, the topological nature of the observed superconductivity, if any, may manifest itself at lower temperatures.

## Results

### Calculating band structures and spin textures at the surface.
We first examine the spin textures at the surface of $\beta$-PdBi$_2$ by first-principles calculations within the framework of the density functional theory. The surface exposed after cleaving a single crystal along (001) plane is expected to be composed of Bi atoms since the weakest bond in the crystal structure is the van der Waals bond between adjacent Bi layers (Fig. 1a inset). Therefore, we mainly focus on low-energy electronic states composed of Bi 6p orbitals at the top-most layer predominating the tunnelling current. (Details of the calculation is described in Methods section.)

The calculated band dispersions and the constant energy contours (CECs) at energy $E = +30$ meV are shown in Fig. 1a, b, respectively. Each of the surface and bulk states is labelled in the same manner as the previous report[20]. The surface states are found in the close vicinity of the bulk bands. The trivial surface state S1 is located near the bulk $\beta$ band, whereas the TSS S2 is located at the bottom of the bulk $\gamma$ and $\delta$ bands. For comparison with the experimental results, we performed a Wannier transformation as written in Methods section. Figure 1c shows the spectral function at +30 meV with the Wannier transformation. The spectral weights of bulk $\gamma$ and $\delta$ bands are very small. For the TSS S2 surrounding $\overline{M}$ points, the weight is negligibly small except for the section parallel to the $\overline{M}-\overline{X}$ direction.

Figure 1d–f show $x$, $y$ and $z$ components of the spin polarisations weighted by the spectral function. It is clear that the spin polarisations are predominantly aligned in-plane, as expected from the symmetry of the crystal structure. It should be noted that not only the TSS S2 but also S1, and even the bulk bands are all spin polarised. The spin orientations of S1 and S2 are consistent with the results of spin-resolved ARPES[20]. The spin polarisations of the bulk bands are likely to be induced by local broken inversion symmetry at the Bi site in the bulk as well as at the surface[24] (Fig. 1a inset). The spin textures of these bulk bands

have never been studied before and play an important role in QPI as we shall show below.

**Spin textures revealed by QPI imaging.** In order to confirm the calculated spin textures experimentally, we utilise QPI imaging. The QPI effect is nothing but the formation of electronic standing waves. In the presence of elastic scatterers, two quantum states on the same CEC with a scattering wave vector $\mathbf{q}$ may interfere and generate a standing wave with a wavelength of $2\pi/|\mathbf{q}|$. This yields $E$-dependent spatial oscillations in the local density-of-states distribution that manifest themselves in the tunnelling conductance $dI/dV|_{V=E/e}$ images, where $I$ is the tunnelling current, $V$ is the bias voltage and $e$ is the elementary charge. Fourier analysis allows us to identify various QPI signals appearing at different $\mathbf{q}$ vectors. The pre-requisite to have QPI signals at $\mathbf{q}$ is that $\mathbf{q}$ must connect two states with high enough spectral weights. In addition, the spin polarisations of these two states should not be anti-parallel because spin-antiparallel states are orthogonal and thereby cannot interfere with each other. In this way, the QPI imaging can provide information on the spin textures in the $\mathbf{k}$ space[25, 26].

Here we outline expected QPI patterns from the calculated spin textures at the surface (Fig. 1d–f). In the case of non-magnetic scatterers, any intra-band back-scattering of spin-polarised bands is forbidden because the spin polarisations of the two states relevant for scattering are always antiparallel due to time-reversal

symmetry. The inter-band back-scatterings from S1 to $\overline{\Gamma}$-centred S2 and from $\alpha$ to $\overline{\Gamma}$-centred S2 are also forbidden, because the spin orientations of S1, S2 and $\alpha$ are the same in the $\overline{\Gamma}$–$\overline{M}$ direction. The inter-band forward-scatterings from S1 to $\overline{\Gamma}$-centred S2 and from S1 to $\alpha$ are allowed but the expected $\mathbf{q}$ vectors are small; it may be difficult to distinguish these QPI signals from extrinsic small $\mathbf{q}$ modulations associated with inhomogeneously distributed unavoidable defects in a real material. As a result, three scattering channels may govern the QPI patterns; inter-band forward-scattering from $\alpha$ to $\overline{\Gamma}$-centred S2 ($\mathbf{q}_1$), inter-band back-scatterings from S1 to $\beta$ band ($\mathbf{q}_2$) and from $\overline{\Gamma}$-centred S2 to $\beta$ band ($\mathbf{q}_3$) (Fig. 1c).

We start our experiments by checking the quality of the cleaved surface using constant-current STM imaging. An atomically flat area as large as $100 \times 100$ nm$^2$ is observed as shown in Fig. 2a. In a magnified image (Fig. 2a inset), at least two kinds of defects are observed as a local suppression and a subtle protrusion surrounded by a suppressed area. Although the exact nature of these defects is unknown, they may work as quasiparticle scatterers that cause QPI. Fourier-transformed STM image (Supplementary Fig. 1) exhibits Bragg peaks that are consistent with the in-plane lattice constant $a_0 = 0.337$ nm[19]. This allows us to identify the crystallographic directions in real and reciprocal spaces.

QPI imaging has been performed in the same field of view of Fig. 2a. Figure 2b, c shows a $dI/dV$ map at $E = +30$ meV and its Fourier-transformed image, respectively. We identify three

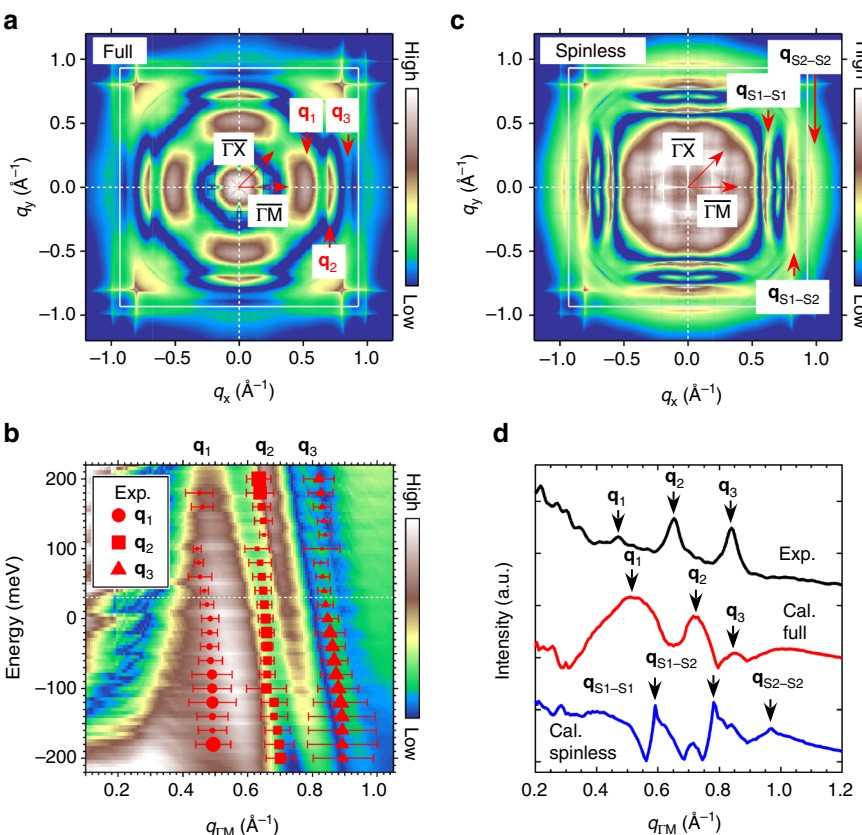

**Fig. 3** Simulated QPI patterns of $\beta$-PdBi$_2$ surface. **a** Simulated QPI pattern at $E = +30$ meV with taking spin degrees of freedom into account (full simulation). The locations of $\mathbf{q}_1$, $\mathbf{q}_2$ and $\mathbf{q}_3$ are shown by arrows. A white square denotes the first Brillouin zone. **b** Simulated energy dispersions of $\mathbf{q}_1$, $\mathbf{q}_2$ and $\mathbf{q}_3$ along $\overline{\Gamma}$–$\overline{M}$ direction with experimental data shown by red symbols. The size of the symbol denotes the signal intensity estimated by Lorenzian fitting. Error bars indicate the full width at half maximum of the fitted peak. **c** Simulated QPI pattern at $E = +30$ meV in which spin degrees of freedom is hypothetically suppressed (spinless simulation). The intra-band back-scatterings in S1 ($\mathbf{q}_{S1-S1}$) and in $\overline{\Gamma}$-centred S2 ($\mathbf{q}_{S2-S2}$), the inter-band back-scattering from S1 to $\overline{\Gamma}$-centred S2 ($\mathbf{q}_{S1-S2}$) appear. **d** Line profiles of **a**, **c** along the $\overline{\Gamma}$–$\overline{M}$ direction. For comparison, corresponding experimental data are also plotted. Each line is offset vertically for clarify

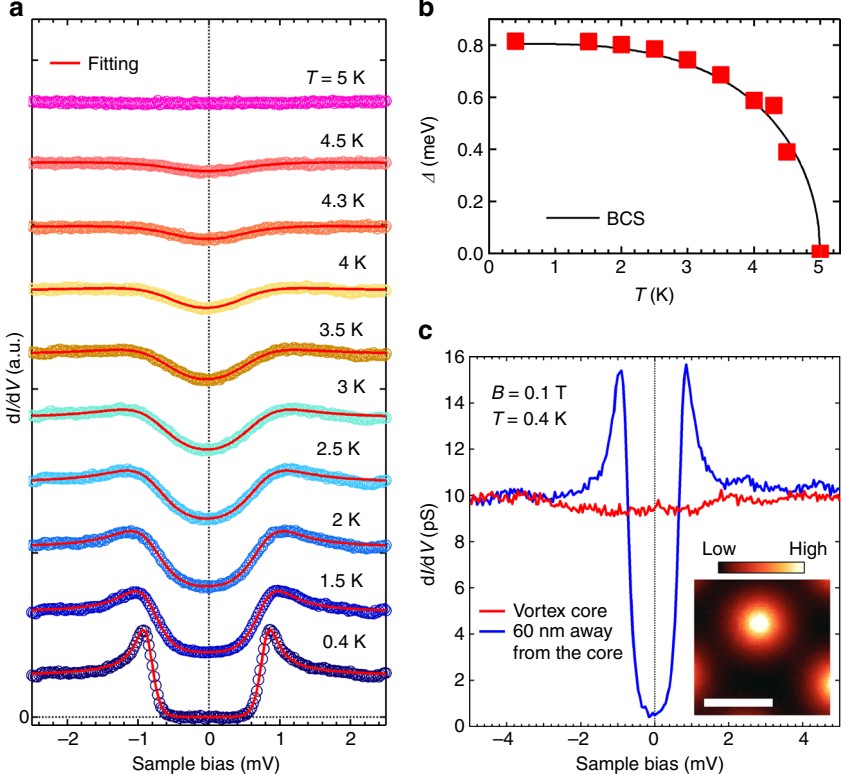

**Fig. 4** Superconducting states at the surface. **a** $dI/dV$ spectrum as a function of temperature. Set-up conditions were $V = +10$ mV, $I = 200$ pA. Bias modulation amplitude $V_{r.m.s.}$ was 18 µV. Solid lines are fitted Dynes functions with thermal broadening taking into account. For clarity, each spectrum is shifted vertically. **b** Temperature dependence of the superconducting gap $\Delta$. A solid line denotes the weak-coupling BCS behaviour. **c** $dI/dV$ spectrum at the centre of vortex core. $T = 0.4$ K and magnetic field $B = 0.1$ T. For comparison, $dI/dV$ taken at 60 nm away from the core is also plotted. Set-up conditions: $V = +10$ mV, $I = 100$ pA. $V_{r.m.s.} = 35$ µV. $dI/dV$ map at $V = 0$ mV is shown in the inset. Set-up conditions: $V = +10$ mV, $I = 100$ pA. $V_{r.m.s.} = 350$ µV. The scale bar corresponds to 100 nm

distinct line-like QPI signals in Fig. 2c, which are parallel with each other and normal to the $\overline{\Gamma}-\overline{M}$ direction. These three QPI signals show hole-like dispersions ($d|\mathbf{q}|/dV_s < 0$) as seen in Fig. 2d. The line-like shapes and hole-like dispersions are consistent with the square-shaped CECs of S1, S2, $\alpha$ and $\beta$ bands shown in Fig. 1b and the band dispersions shown in Fig. 1a, respectively. Namely, the observed QPI signals may correspond to $\mathbf{q}_1$, $\mathbf{q}_2$ and $\mathbf{q}_3$.

We confirm this conjecture by comparing experimental data with the numerically simulated QPI patterns calculated by using the results of first-principles calculations with the standard $T$-matrix formalism. In addition to a full (standard) simulation, we have also performed a spinless simulation. The spin degree of freedom is included in the former and hypothetically suppressed in the latter. The contrast between the two simulations highlights the role of the spin textures. (See Methods section for details.)

The result of the full simulation at $E = +30$ meV is shown in Fig. 3a. Three parallel line-like QPI signals normal to the $\overline{\Gamma}-\overline{M}$ direction are identified. These features correspond to $\mathbf{q}_1$, $\mathbf{q}_2$ and $\mathbf{q}_3$ indicated in Fig. 1c and their energy evolutions reasonably agree with the experimental observation (Fig. 3b), indicating that $\mathbf{q}_1$, $\mathbf{q}_2$ and $\mathbf{q}_3$ scatterings indeed dominate the QPI patterns.

The key role of the spin textures becomes evident if we compare the result of full simulation with that of spinless simulation (Fig. 3c), and the experimental data. Figure 3d compares line profiles along the $\overline{\Gamma}-\overline{M}$ direction from experimental and simulated QPI patterns. In the spinless simulation, there are three sharp peaks but their $|\mathbf{q}|$ values do not agree with the experimental observations. These sharp peaks correspond to the intra-surface-state back-scatterings in S1 ($\mathbf{q}_{S1-S1}$) and in $\overline{\Gamma}$-centred S2 ($\mathbf{q}_{S2-S2}$), and inter-band back-scattering from S1 to

$\overline{\Gamma}$-centred S2 ($\mathbf{q}_{S1-S2}$). Because the spin orientations are antiparallel between initial and final states of these scatterings, they are suppressed in the full simulation that well captures the experimental observations.

The above comprehensive approach combining experiments and calculations provide the following important implications. First, our SI-STM data primarily reflect the electronic state of the top-most Bi layer as we assumed. Second, the spin degrees of freedom is crucial for QPI. Finally, the calculated spin textures shown in Fig. 1d–f adequately capture the real spin textures at the surface. Notably, the fourfold QPI pattern is similarly identified at $E_F$ in the normal state ($T = 1.5$ K, $B = 12$ T) as shown in Supplementary Fig. 2. This indicates that the spin textures discussed here indeed exist at $E_F$ in the normal state.

**Superconductivity at the surface.** Given the surface sensitivity of our measurement evidenced in the QPI patterns, we are able to argue the nature of superconductivity at the surface. Figure 4a shows the temperature $T$ evolution of the tunnelling spectrum in the SC state. At $T = 0.4$ K, the SC gap fully opens and there is no residual spectral weight inside the gap, indicating that all of the states at the surface are gapped. Each spectrum can be fitted well with the Dynes function for the single isotropic gap, meaning that the SC gap is $\mathbf{k}$ independent. We also investigate QPI patterns near the SC gap energy and find no clear superconductivity-induced QPI signals except at $\mathbf{q} = 0$ (Supplementary Fig. 2), which means that the SC gap is spatially uniform. The SC gap amplitude $\Delta(T)$ is estimated to be 0.8 meV at $T = 0.4$ K, being reasonably consistent with the result of air-cleaved sample[21].

We note that any noticeable spectroscopic features are not observed near $E_F$ at step edges (Supplementary Fig. 3). The temperature dependence of the SC gap well follows the BCS behaviour (Fig. 4b).

We next examine the electronic state of vortices that may host Majorana fermions[6, 27, 28]. Isotropic vortices are clearly imaged in the d$I$/d$V$ map at $E = 0$ meV (Fig. 4c inset), as being similar to the previous result obtained at the surface of the air-cleaved bulk crystal[21]. The line profile of the vortex core is also reasonably consistent with the previous result[29] (Supplementary Fig. 4). To estimate the in-plane coherence length and compare it with that obtained from $H_{c2}$ measurements, detailed magnetic field dependence of the vortex core size is required as proposed recently[29]. Considering the good agreement with our result and those in ref. [29], we believe that a similar coherence length as discussed in refs. [21, 29] would be expected in our samples. The spectrum taken in the vortex core is almost flat (Fig. 4c). This result indicates that the core is in the dirty limit (mean free path $l < \xi$) where vortex bound states as well as the Majorana zero mode are not well defined. Although the zero-energy local density-of-states peak in the vortex core cannot be an evidence of the Majorana zero mode by itself[18], it would be possible to investigate the spin structures that are unique to the Majorana state[27, 28], for example. We anticipate that $\beta$-PdBi$_2$ will be a good touchstone for these theories in future because it is a stoichiometric material that can be made cleaner in principle.

## Discussion

Our concurrent investigations of the surface electronic states and superconductivity reveal that the SC gap fully opens in all of the spin-polarised surface states. Since the mixing of spin-singlet and triplet order parameters is generally expected in the presence of non-degenerate spin-polarised surface states[1, 2] and the triplet component can possess nodes, it is intriguing to argue the possible nodal structure of the SC states of $\beta$-PdBi$_2$. It has been shown that the spin-triplet order parameter $\mathbf{d}(\mathbf{k})$ favours to be aligned along the spin direction when space inversion symmetry is broken[30]. If this is the case at the surface of $\beta$-PdBi$_2$ where the point group symmetry is lowered from $D_{4h}$ to $C_{4v}$, the triplet pairing is of $p$-wave type with nodes along the out-of-plane direction[30]. Since the surface states are two dimensional in nature, the out-of-plane nodes would not be active and the SC gap should look like isotropic as observed.

Next we discuss the amplitude of the SC gap. The SC gap amplitude in the presence of the parity mixing is given by $|\Psi_s(\mathbf{k})| \pm |\mathbf{d}(\mathbf{k})|$ where $\Psi_s$ denotes $s$-wave order parameter that represents the bulk gap[2, 30–32]. Thus the difference between the SC gap amplitudes of bulk and surface gives us an estimate of the amplitude of the $p$-wave component. The bulk SC gap has been estimated to be about 0.9 meV by the specific heat measurement[23], which is close to the value of 0.8 meV that we observed at the surface. According to the theory[32], the difference between the surface SC gap and the bulk counterpart depends on how far the surface state is separated from the bulk band at $E_F$ in $\mathbf{k}$ space. We estimate the quantity $\delta \equiv \left(k_F^D - k_F\right)/k_F$, which is introduced in ref. [32], to be about 0.05 ($k_F^D$ and $k_F$ denote the Fermi momenta of the TSS and the bulk band, respectively). This is reasonably small to explain the similar SC gap amplitudes between the surface and the bulk. Namely, even though it is allowed, the $p$-wave component at the surface of $\beta$-PdBi$_2$ is small. The absence of edge states shown in Supplementary Fig. 3 might be due to the small $p$-wave component. In order to detect this small $p$-wave component, if any, we need to avoid comparing the results of two different experiments, STM and specific heat, as well as to improve the energy resolution. As the bulk bands appear at the surface, too,

it would be possible to detect the surface and bulk SC gaps simultaneously by STM alone, provided higher energy resolution could be achieved. To this end, the use of a SC tip at lower temperatures is important.

## Methods

**Sample preparation and STM measurements.** The $\beta$-PdBi$_2$ single crystals were grown by a melt growth method[20], and characterised by x-ray diffraction (XRD) and transport measurements. Pd (3N5) and Bi (5 N) at a molar ratio of 1:2 were encapsulated in an evacuated quartz tube, pre-reacted at temperature above 1000 °C until it completely melted and mixed. Then, it was kept at 900 °C for 20 h, cooled down at a rate of 3 °C/h down to 500 °C, and finally rapidly quenched into cold water. PdBi$_2$ has two different crystallographic and superconducting phases: $\alpha$-phase with the space group C2/m ($T_c \sim 1.7$ K[33]) and $\beta$-phase with I4/mmm ($T_c \sim 5.4$ K). The last quenching procedure is important to selectively obtain the $\beta$-phase. Any trace amount of $\alpha$-phase was not detected by XRD. For STM measurements, we used single crystals with residual resistivity ratio of 15, larger than those of the previous reports[19, 21, 23]. The crystals were cleaved along the (001) plane at room temperature in ultra high vacuum conditions to obtain clean and flat surfaces needed for STM. A commercial $^3$He-based STM system (UNISOKU USM-1300) modified by ourselves[34] was employed in this study. We used electrochemically etched tungsten tips, which were cleaned and sharpened by field ion microscopy. The tips were subsequently treated and calibrated on clean Au(100) surfaces before used for $\beta$-PdBi$_2$. We applied bias voltages to the sample whereas the tip was virtually grounded at the current-voltage converter (Femto LCA-1K-5G). Tunnelling spectra were measured by the software-based lock-in detector included in the commercial STM control system (Nanonis).

**Calculation of the electronic state.** To calculate the surface electronic structure and its corresponding QPI pattern, we first performed a DFT calculation using the Perdew–Burke–Ernzerhof exchange-correlation functional[35] as implemented in the WIEN2K program[36]. Relativistic effects including spin-orbit coupling were fully taken into account. For all atoms, the muffin-tin radius $R_{MT}$ was chosen such that its product with the maximum modulus of reciprocal vectors $K_{max}$ become $R_{MT}K_{max} = 7.0$. Considering the tetragonal phase of $\beta$-PdBi$_2$, the corresponding Brillouin zone was sampled using a $20 \times 20 \times 5$ $k$-mesh. For the surface calculations, a 100 unit tight-binding supercell was constructed using maximally localised Wannier functions[37–39]. The 6p orbitals of Bi atoms were chosen as the projection centres.

**QPI simulation.** We simulated QPI patterns by incorporating the eigenvalues and eigenvectors obtained from our tight-binding supercell calculations into the standard $T$-matrix formalism[40]. The top-most Bi 6p orbitals were considered in the calculations. For all the simulations, the broadening factor was chosen to be 5 meV, and a localised, spin-preserving and orbital-preserving scatterer of strength 0.1 eV were employed. For comparison with the experimental results, we performed a basis transformation from the lattice model to the continuum model with the Wannier function[41] constructed by projecting the Bi 6p orbitals. We calculated the local density of state at 0.5 nm above the top-most Bi atoms. The basis transformation is also applied to the spectral function shown in Fig. 1c to present the electronic structure used for the QPI calculation. For the spinless simulation, the sum of the Green's functions constructed from the spin-up and spin-down subspaces were used for the $T$-matrix calculations.

**Data availability.** All relevant data are available on request, which should be addressed to K.I.

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

# ARTICLE

8. Levy, N. et al. Experimental evidence for s-wave pairing symmetry in superconducting $Cu_xBi_2Se_3$ single crystals using a scanning tunneling microscope. *Phys. Rev. Lett.* **110**, 117001 (2013).

9. Du, G. et al. Drive the Dirac electrons into cooper pairs in $Sr_xBi_2Se_3$. *Nat. Commun.* **8**, 14466 (2017).

10. Wang, E. et al. Fully gapped topological surface states in $Bi_2Se_3$ films induced by a d-wave high-temperature superconductor. *Nat. Phys.* **9**, 621–625 (2013).

11. Wang, M. X. et al. The coexistence of superconductivity and topological order in the $Bi_2Se_3$ thin films. *Science* **336**, 52–55 (2012).

12. Xu, S. Y. et al. Momentum-space imaging of cooper pairing in a half-Dirac-gas topological superconductor. *Nat. Phys.* **10**, 943–950 (2014).

13. Xu, J. P. et al. Artificial topological superconductor by the proximity effect. *Phys. Rev. Lett.* **112**, 217001 (2014).

14. Xu, J. P. et al. Experimental detection of a Majorana mode in the core of a magnetic vortex inside a topological insulator-superconductor $Bi_2Te_3/NbSe_2$ Heterostructure. *Phys. Rev. Lett.* **114**, 017001 (2015).

15. Nadj-Perge, S. et al. Observation of Majorana fermions in ferromagnetic atomic chains on a superconductor. *Science* **346**, 602–607 (2014).

16. Mourik, S. et al. Signatures of Majorana fermions in hybrid superconductor-semiconductor nanowire devices. *Science* **336**, 1003–1007 (2012).

17. Ali, M. N., Gibson, Q. D., Klimczuk, T. & Cava, R. J. Noncentrosymmetric superconductor with a bulk three-dimensional Dirac cone gapped by strong spin-orbit coupling. *Phys. Rev. B* **89**, 020505(R) (2014).

18. Guan, S. Y. et al. Superconducting topological surface states in the noncentrosymmetric bulk superconductor $PbTaSe_2$. *Sci. Adv.* **2**, e1600894 (2016).

19. Imai, Y. et al. Superconductivity at 5.4 K in $\beta$-$Bi_2Pd$. *J. Phys. Soc. Jpn.* **81**, 113708 (2012).

20. Sakano, M. et al. Topologically protected surface states in a centrosymmetric superconductor $\beta$-$PdBi_2$. *Nat. Commun.* **6**, 8595 (2015).

21. Herrera, E. et al. Magnetic field dependence of the density of states in the multiband superconductor $\beta$-$Bi_2Pd$. *Phys. Rev. B* **92**, 054507 (2015).

22. Lv, Y. F. et al. Experimental signature of topological superconductivity and Majorana zero modes on $\beta$-$Bi_2Pd$ thin films. *Sci. Bull.* **62**, 852–856 (2017).

23. Kačmarčík, J. et al. Single-gap superconductivity in $\beta$-$Bi_2Pd$. *Phys. Rev. B* **93**, 144502 (2016).

24. Zhang, X., Liu, Q., Luo, J. W., Freeman, A. J. & Zunger, A. Hidden spin polarization in inversion-symmetric bulk crystals. *Nat. Phys.* **10**, 387–393 (2014).

25. Pascual, J. I. et al. Role of spin in quasiparticle interference. *Phys. Rev. Lett.* **93**, 196802 (2004).

26. Roushan, P. et al. Topological surface states protected from backscattering by chiral spin texture. *Nature* **460**, 1106–1109 (2009).

27. Nagai, Y., Nakamura, H. & Machida, M. Spin-polarized Majorana bound states inside a vortex core in topological superconductors. *J. Phys. Soc. Jpn* **83**, 064703 (2014).

28. Kawakami, T. & Hu, X. Evolution of density of states and a spin-resolved checkerboard-type pattern associated with the Majorana bound state. *Phys. Rev. Lett.* **115**, 177001 (2015).

29. Fente, A. et al. Field dependence of the vortex core size probed by scanning tunneling microscopy. *Phys. Rev. B* **94**, 014517 (2016).

30. Frigeri, P. A., Agterberg, D. F., Koga, A. & Sigrist, M. Superconductivity without inversion symmetry: MnSi versus $CePt_3Si$. *Phys. Rev. Lett.* **92**, 097001 (2004).

31. Hayashi, N., Wakabayashi, K., Frigeri, P. A. & Sigrist, M. Temperature dependence of the superfluid density in a noncentrosymmetric superconductor. *Phys. Rev. B* **73**, 024504 (2006).

32. Mizushima, T., Yamakage, A., Sato, M. & Tanaka, Y. Dirac-fermion-induced parity mixing in superconducting topological insulators. *Phys. Rev. B* **90**, 184516 (2014).

33. Mitra, S. et al. Probing the superconducting gap symmetry of $\alpha$-$PdBi_2$: a penetration depth study. *Phys. Rev. B* **95**, 134519 (2017).

34. Hanaguri, T. Development of high-field STM and its application to the study on magnetically-tuned criticality in $Sr_3Ru_2O_7$. *J. Phys.: Conf. Ser.* **51**, 514–521 (2006).

35. Perdew, J. P., Burke, K. & Ernzerhof, M. Generalized gradient approximation made simple. *Phys. Rev. Lett.* **77**, 3865–3868 (1996).

36. Balha, P., Schwarz, K., Madsen, G., Kvasnicka, D. & Luitz, J. WIEN2K package, Version 13.1 (2013).

37. Souza, I., Marzari, N. & Vanderbilt, D. Maximally localized Wannier functions for entangled energy bands. *Phys. Rev. B* **65**, 035109 (2001).

38. Mostofi, A. A. et al. Wannier90: a tool for obtaining maximally localised Wannier functions. *Comp. Phys. Commun.* **178**, 685–699 (2008).

39. Kunes, J. et al. Wien2wannier: from linearized augmented plane waves to maximally localized Wannier functions. *Comp. Phys. Commun.* **181**, 1888–1895 (2010).

40. Kohsaka, Y. et al. Spin-orbit scattering visualized in quasiparticle interference. *Phys. Rev. B* **95**, 115307 (2017).

41. Kreisel, A. et al. Interpretation of scanning tunneling quasiparticle interference and impurity states in cuprates. *Phys. Rev. Lett.* **114**, 217002 (2015).

## Acknowledgements

The authors thank M. Sakano, K. Ishizaka, T. Mizushima, Y. Tanaka and Y. Yanase for valuable discussions and Y. Okada for experimental advices. This work was supported by a CREST project (JPMJCR16F2) from Japan Science and Technology Agency (JST) and JSPS KAKENHI Grant Nos.16K05465, 16H06012, and 16H03847. M.S.B. was supported by CREST, JST (Grant No. JPMJCR16F1).

## Author contributions

K.I. carried out the experiments and the data analyses with assistance from Y.K., T.M. and T.H. Y.K., T.S., and M.S.B. carried out the first-principles calculations and numerical simulations. K.O. and T.S. synthesised and characterised the single crystals. T.H. supervised the project. K.I., Y.K. and T.H. wrote the manuscript.

## Additional information

**Competing interests:** The authors declare no competing financial interest.

**Change history:** A correction to this article has been published and is linked from the HTML version of this paper.

