## [Peer Review File · Nature Communications]

Reviewers' comments:

Reviewer #1 (Remarks to the Author):

Authors make a detailed study about the bandstructure of the new material beta-Bi₂Pd and try to interpret their results in the light of anomalous properties appearing at the surface. Authors take as face value "facts", such as "given the fact that the electronic states detected by SI-STM are all spin-polarized, we can safely conclude that the spin-polarized states exhibit fully gapped superconductivity". This and similar statements are, in my view, not really justified.

It is also very confusing that authors report the constant energy contours at zero energy. At zero meV, the superconducting gap is open and the constant energy contours should be fully absent. May be authors can write ≈ 0 meV and say in the caption that the superconducting gap opening was not studied. For the same reason, it makes no sense to discuss experiments made at zero energy. The bias voltage modulation was probably much larger than the gap size, so this is certainly not zero energy.

In all, I think that authors have searched to find something "unusual" or "spectacular" in the band dispersion they observe. In this process, however, they seem to have lost track of what their experiment is telling them. Their experiment is interesting and, if discussed within the nice physics that can be found in this system and with more measurements close to the superconducting gap, it might lead to a publication that is suitable for a wide audience, as aimed by nat com. As it is now, and without further data, I am convinced it should be published in a specialized venue.

Several details that could be of help in re-writing the paper:

The discussion about Majorana physics in the vortex core is quite misplaced. I would strongly advice to remove it or to explain why and under which conditions such physics might arise in this superconductor.

It is a pity that authors state that the superconducting gap is k independent but provide no data showing the QPI signal within the superconducting gap. To ascertain that the gap is indeed k-independent, they should make such a measurement. If this is now unavailable, they should at least place an argument, maybe using the shape of vortex cores, and mention this. Also, they cite two publications. In one of these publications, the vortex cores are round and in the other the vortex cores have a square shape. If the gap is indeed isotropic within the strongly fourfold bandstructure found by the authors, this is quite remarkable in my view and should be discussed.

No details are provided for the crystal growth. This is important, the phase is obtained by quenching, introducing naturally defects in it. All other papers in the subject provided such an information.

There is clearly additional dispersion close to the zone center that is not taken into account at all in the data. Apparently, around 100 meV, the dispersion vanishes. This seems quite amazing to me and I do not understand why would that happen ?

Reviewer #2 (Remarks to the Author):

The authors present a detail STM and DFT study on PdBi₂ single crystals. They demonstrate the spin structure on both the surface bands and the bulk bands are spin-polarized and in-plane at Fermi level by using QPI and DFT (which is novel compared with the previous STM studies). The superconducting tunneling spectrum at very low temperature shows a full superconducting gap, which means all spin-polarized bands resolved by QPI are fully gapped. These two related phenomena can lead to topological superconductivity at the surface, which is an important subject

in condensed matter physics. Although the bound state at vortex core is absent in STM data due to strong impurity scattering and the spectroscopic signature of topological superconductivity on the surface state is weak due to the energy resolution (and low $T_c \sim 5.4\text{K}$), the stoichiometric materials with these properties are extremely rare and I believe the findings here can generate further investigation in these interesting materials. Therefore, the work presented here is definitely an exciting subject for broad readership. Their data and the analysis are of high quality and I highly recommend this manuscript to be published in Nature Communication with few comments and suggestions below.

(1) In Fig.2, the dI/dV and QPI image at $E=0$ are measured in the superconducting state at $T=1.5\text{K}$. Since it is fully gapped, one should not see any QPI at $E=0\text{meV}$. The observation here is because the modulation amplitude is larger than the gap. The authors should make it clear (at least in the caption) to avoid any confusion.

(2) As the authors mention in the last paragraph of discussion, it would be possible to detect the surface and bulk SC gaps simultaneously by STM alone. Since the authors have the capability to perform QPI measurements at much lower temperature (say $T=0.4\text{K}$), it will be very interesting to see if they can find any signature of p-wave pairing.

(3) The absence of zero energy bound state in a vortex core is attributed to the impurity scattering, which the authors call for cleaner samples. This statement is definitely true. At the same time, multiple spin-polarized bands are present at Fermi level so the absence of zero bias peak in the tunneling spectrum may also be due to the instability from the scattering between Majorana zero modes.

(4) If the topological superconductivity exists in PdBi_2 , it's like a 2D case so an edge state is expected. This is absent in STM data. In fact, to the best of my knowledge, such predicted edge state has not been observed in any reported TSC candidate materials, which is very puzzling.

Reply to the Reviewer #1:

We thank the reviewer for making useful comments to improve the manuscript.

Authors make a detailed study about the bandstructure of the new material β -Bi₂Pd and try to interpret their results in the light of anomalous properties appearing at the surface. Authors take as face value “facts”, such as “given the fact that the electronic states detected by SI-STM are all spin-polarized, we can safely conclude that the spin-polarized states exhibit fully gapped superconductivity”. This and similar statements are, in my view, not really justified.

We agree that the word “fact” may not be appropriate. Rather, this denotes a logical consequence deduced from our experiments, calculations and previous ARPES results. We amended the corresponding part in the revised manuscript. We believe that there is no other logical flaw in the rest of the manuscript.

It is also very confusing that authors report the constant energy contours at zero energy. At zero meV, the superconducting gap is open and the constant energy contours should be fully absent. May be authors can write ≈ 0 meV and say in the caption that the superconducting gap opening was not studied. For the same reason, it makes no sense to discuss experiments made at zero energy. The bias voltage modulation was probably much larger than the gap size, so this is certainly not zero energy.

We agree that the “QPI image at 0 meV” was confusing. As the reviewer correctly pointed out, the QPI signal at 0 meV shown in the previous manuscript is caused by the large modulation amplitude. To avoid the confusion, we replaced the QPI images at 0 meV with those at +30 meV in Figs. 2(b) and (c) and added a sentence about the lock-in modulation amplitude in the figure caption. Accordingly, the constant energy contours, the spectral function and the spin textures at +30 meV are shown in Fig. 1 in the revised manuscript.

In all, I think that authors have searched to find something “unusual” or “spectacular” in the band dispersion they observe. In this process, however, they seem to have lost track of what their experiment is telling them.

We apologize that the message of our work was not clear in the previous manuscript. The interesting point that we revealed in this work is that all of the Fermi surfaces at the surface of β -PdBi₂ are *spin-polarized* and all of them exhibit *fully-gapped* superconductivity. In order to make this point clearer, we have slightly changed the title.

Their experiment is interesting and, if discussed within the nice physics that can be found in this system and with more measurements close to the superconducting gap, it might lead to a publication that is suitable for a wide audience, as aimed by nat com.

We believe that we performed our experiments and analyses within the standard frameworks but it is true that we did not discuss the QPI images in the superconducting-gap energy scale, even though we already had data. This is because the

data do not exhibit noticeable \mathbf{q} -space features in this energy scale. Nevertheless, we agree with the reviewer that the QPI data at low energies represent the different aspect of superconductivity. Therefore, we added a new supplementary figure showing QPI images at -1 meV, which is near the gap-edge energy, in the superconducting state ($T = 1.5$ K, $B = 0$ T) and in the normal state ($T = 1.5$ K, $B = 12$ T). We did not find any superconductivity-induced QPI signals at finite \mathbf{q} , which means that the superconducting gap is spatially uniform. We also added a QPI image at 0 meV in the normal state ($T = 1.5$ K, $B = 12$ T). The image shows the characteristic four-fold QPI pattern, meaning that the spin-polarized states discussed in this study surely exist at 0 meV in the normal state.

The discussion about Majorana physics in the vortex core is quite misplaced. I would strongly advice to remove it or to explain why and under which conditions such physics might arise in this superconductor.

Possible Majorana states in the vortex core are of current interest in the community and thus we believe that it is reasonable to mention it even though we did not find any evidence of it so far. As mentioned by the reviewer #2, the absence of the features in the vortex core itself may promote a new idea. Another message we wish to deliver is that, since β -PdBi₂ is a stoichiometric material, there is a lot of room for improvement of the sample quality. As far as we understand, the condition to observe the Majorana state in the vortex core has not been established yet but there are a few theoretical proposals to distinguish the Majorana state from the conventional Caroli-de Gennes-Matricon state (Refs. 27, 28). As we described in the manuscript, we anticipate that β -PdBi₂ can be a good touchstone for these theories, provided the sample quality would be improved.

It is a pity that authors state that the superconducting gap is \$\mathbf{k}\$ independent but provide no data showing the QPI signal within the superconducting gap. To ascertain that the gap is indeed \$\mathbf{k}\$ -independent, they should make such a measurement.

We apologize that we omitted the QPI data in the superconducting-gap energy scale, which are now included in the supplementary materials as mentioned above. However, we think that it is difficult to discuss the \mathbf{k} -space structure of the superconducting gap from the Bogoliubov QPI in β -PdBi₂. To observe the Bogoliubov QPI, the superconducting gap should be \mathbf{k} dependent. Our high-resolution point spectra shown in Fig. 4 can perfectly be fitted by the Dynes function for the isotropic gap and there is no detectable spectral weight at the Fermi energy. These observations mean that all of the Fermi surfaces are gapped and all of the \mathbf{k} points possess almost the same gap amplitude. The upper bound of the gap-size distribution roughly corresponds to the thermal broadening energy at 0.4 K ($3.5k_B \times 0.4$ K ~ 0.12 meV) that is much smaller than the estimated gap amplitude of 0.8 meV. We do not exclude the possibility of small but finite \mathbf{k} dependence, but our conclusion, fully gapped superconductivity in the spin-polarized Fermi surfaces, should be robust.

If this is now unavailable, they should at least place an argument, maybe using the shape of vortex cores, and mention this. Also, they cite two publications. In one of these publications, the vortex cores are round and in the other the vortex cores have a square shape. If the gap is indeed isotropic within the strongly fourfold bandstructure found by

the authors, this is quite remarkable in my view and should be discussed.

The reviewer is correct that the shape of the vortex core is determined not only by the gap anisotropy but also by the shape of the Fermi surface. In β -PdBi₂, surface band structure is indeed square shaped but the calculated spectral weight distribution in momentum space is rather complicated as shown below. Therefore, it is difficult to predict the vortex-core shape from intuitive arguments. As for the difference of the vortex-core shapes in the bulk sample and the film, we do not have enough information to mention the exact origin. Since the superconducting-gap spectra are very different between the bulk sample and the film, it may be difficult to compare these two on an equal footing. Considering these situations, we now recognize that the isotropic vortex itself is rather weak to claim the isotropic gap, even though they do not contradict each other. Accordingly, we amended the corresponding part in the revised manuscript, just by pointing out the consistency between our result and the previous result in the bulk sample (Ref. 21).

Figure. Calculated spectral function at $E = 0$ meV with the Wannier transformation. To highlight the largest spectral weights, different colour-scale from that of Fig.1c in the main text is adopted. There exist substantial spectral weights in the $\overline{\Gamma} - \overline{X}$ direction as well as in the $\overline{\Gamma} - \overline{M}$ direction.

No details are provided for the crystal growth. This is important, the phase is obtained by quenching, introducing naturally defects in it. All other papers in the subject provided such an information.

We apologize for the lack of the information. The details about the crystal growth are added in Methods section in the revised manuscript.

There is clearly additional dispersion close to the zone center that is not taken into account at all in the data. Apparently, around 100 meV, the dispersion vanishes. This seems quite amazing to me and I do not understand why would that happen ?

The additional signals around $\mathbf{q} = 0$ are mostly associated with the unavoidable inhomogeneous distribution of defects in a real material and thus extrinsic in origin.

The suppressed QPI signals around +100 meV are due to the so-called set-point effect that inevitably appears in the dI/dV images. In STM measurements, tunnelling current is regulated to be constant by the feedback loop that is opened during the spectroscopic measurement. This forces the integration of the Fourier component at finite \mathbf{q} from Fermi energy to the bias voltage to be zero. Therefore, each Fourier component (except for $\mathbf{q} = 0$ component) should become zero, at least once, between 0 mV and the setup

bias voltage for imaging. In the present case, the setup bias is +200 mV and the zero-crossing occurs around +100 mV. We describe this in the revised Supplementary Information.

Reply to the Reviewer #2:

The authors present a detail STM and DFT study on PdBi₂ single crystals. They demonstrate the spin structure on both the surface bands and the bulk bands are spin-polarized and in-plane at Fermi level by using QPI and DFT (which is novel compared with the previous STM studies). The superconducting tunneling spectrum at very low temperature shows a full superconducting gap, which means all spin-polarized bands resolved by QPI are fully gapped. These two related phenomena can lead to topological superconductivity at the surface, which is an important subject in condensed matter physics. Although the bound state at vortex core is absent in STM data due to strong impurity scattering and the spectroscopic signature of topological superconductivity on the surface state is weak due to the energy resolution (and low $T_c \sim 5.4$ K), the stoichiometric materials with these properties are extremely rare and I believe the findings here can generate further investigation in these interesting materials. Therefore, the work presented here is definitely an exciting subject for broad readership. Their data and the analysis are of high quality and I highly recommend this manuscript to be published in Nature Communication with few comments and suggestions below.

We sincerely thank the reviewer for his/her strong recommendation for publication of our manuscript and appreciate insightful comments.

(1) In Fig.2, the dI/dV and QPI image at $E = 0$ are measured in the superconducting state at $T = 1.5$ K. Since it is fully gapped, one should not see any QPI at $E = 0$ meV. The observation here is because the modulation amplitude is larger than the gap. The authors should make it clear (at least in the caption) to avoid any confusion.

This issue is also raised by the reviewer #1 and we apologize for this confusion. As the reviewer correctly pointed out, the QPI signal at 0 meV shown in the previous manuscript is caused by the large modulation amplitude. In the revised manuscript, all related images are replaced by those at +30 meV and a sentence about the lock-in modulation amplitude is added in the figure caption. We also added normal-state 0 meV data obtained by applying high magnetic field in Supplementary Information.

(2) As the authors mention in the last paragraph of discussion, it would be possible to detect the surface and bulk SC gaps simultaneously by STM alone. Since the authors have the capability to perform QPI measurements at much lower temperature (say $T = 0.4$ K), it will be very interesting to see if they can find any signature of p -wave pairing.

To achieve an enough signal-to-noise ratio within the limited experimental time, QPI measurements generally demand relatively large modulation amplitude for spectroscopy (larger than the thermal broadening energy). Therefore, drastic improvement of energy resolution cannot be expected even if we perform the experiment at 0.4 K. Nevertheless

it should be important to go to lower temperatures for higher-resolution point spectra as we mentioned in the Discussion section.

(3) The absence of zero energy bound state in a vortex core is attributed to the impurity scattering, which the authors call for cleaner samples. This statement is definitely true. At the same time, multiple spin-polarized bands are present at Fermi level so the absence of zero bias peak in the tunneling spectrum may also be due to the instability from the scattering between Majorana zero modes.

We appreciate the reviewer very much for suggesting this very interesting possibility. However, our current data may not be enough to discuss this scenario. We will think of possible experiments that are relevant for this suggestion.

(4) If the topological superconductivity exists in PdBi₂, it's like a 2D case so an edge state is expected. This is absent in STM data. In fact, to the best of my knowledge, such predicted edge state has not been observed in any reported TSC candidate materials, which is very puzzling.

The absence of the edge states might be due to small p -wave components in the candidate materials. To detect such edge states, we need to find a material which possesses a substantial p -wave component. We added this speculation in the revised manuscript.

REVIEWERS' COMMENTS:

Reviewer #1 (Remarks to the Author):

Authors have taken carefully into account all suggestions and answered to all comments. In particular, I now understand where the 100 mV feature comes (although one serious point requires correction before publication, see below) and this brings the whole rest of the work into the right place to me. Overall, I do not necessarily agree with each one and every statement given by authors. But authors view's are sound and relevant to me, and are also very well discussed in their text and in their answers. I therefore have now no doubt to recommend this paper for publication in nature communications.

Still, there must be an error somewhere and authors should correct for it. In one figure, they state that the setpoint is at 200 mV but the feature is located at 100 mV. How can that be ? Authors should certainly make this point clear before publication. From the rest of the paper I guess it's a simple error, but this is the authors' job.

The result of having an isotropic gap and close-by probably spin polarized bands is interesting, important and should be communicated to an ample audience.

As a comment, authors provide a measurement of the vortex core size and I wish to draw attention to this paper, PHYSICAL REVIEW B 94, 014517 (2016). They can use the equations provided in that paper and obtain a much better agreement between ξ and their vortex core profile. The equation they use is just a phenomenological approach and it is hardly to be said coming from GL theory.

Reviewer #2 (Remarks to the Author):

Overall, I am satisfied with the authors' response to my previous comments. I also found their response to Reviewer 1's comments reasonable. I believe these observations will inspire future theoretical and experimental studies on both topological superconductivity and Majorana zero modes in similar stoichiometric materials. Thus, I highly recommend this paper for publication in Nature Communications.

Reply to the Reviewer #1:

Authors have taken carefully into account all suggestions and answered to all comments. In particular, I now understand where the 100 mV feature comes (although one serious point requires correction before publication, see below) and this brings the whole rest of the work into the right place to me. Overall, I do not necessarily agree with each one and every statement given by authors. But authors view's are sound and relevant to me, and are also very well discussed in their text and in their answers. I therefore have now no doubt to recommend this paper for publication in nature communications.

We sincerely thank the reviewer for his/her strong recommendation for publication of our manuscript.

Still, there must be an error somewhere and authors should correct for it. In one figure, they state that the setpoint is at 200 mV but the feature is located at 100 mV. How can that be ? Authors should certainly make this point clear before publication. From the rest of the paper I guess it's a simple error, but this is the authors' job.

The suppression of QPI signals should occur at any bias voltages between 0 and the set-point bias voltage as expected from Supplementary Eq. (4). In our experiment, the set-point bias voltage is +200 mV. Thus the QPI suppression appeared at +100 mV is reasonable.

The result of having an isotropic gap and close-by probably spin polarized bands is interesting, important and should be communicated to an ample audience.

We thank the referee for recognizing the significance of our findings.

As a comment, authors provide a measurement of the vortex core size and I wish to draw attention to this paper, PHYSICAL REVIEW B 94, 014517 (2016). They can use the equations provided in that paper and obtain a much better agreement between ξ and their vortex core profile. The equation they use is just a phenomenological approach and it is hardly to be said coming from GL theory.

We appreciate the reviewer for suggesting a new analysis method and also agree that the equation in the previous manuscript is based on just a phenomenological approach although it has been conventionally used by many groups. To estimate the coherence length from the vortex core profile, (if we understood the suggested paper correctly) we need detailed magnetic field dependence of the vortex core size, which is currently not available for us. Hence, in the revised manuscript, we just show a good agreement of the line profiles of the vortex cores between our sample and that of the suggested paper to claim that the coherence length in our UHV-cleaved sample is similar to that of the air-cleaved sample.

Reply to the Reviewer #2:

Overall, I am satisfied with the authors' response to my previous comments. I also found their response to Reviewer 1's comments reasonable. I believe these observations will inspire future theoretical and experimental studies on both topological superconductivity and Majorana zero modes in similar stoichiometric materials. Thus, I highly recommend this paper for publication in Nature Communications.

We thank the reviewer again for his/her strong recommendation for publication of our manuscript.